# Facile Fabrication of Robust and Fluorine-Free Superhydrophobic PDMS/STA-Coated Cotton Fabric for Highly Efficient Oil-Water Separation

Daibin Tang and Enzhou Liu *

School of Chemical Engineering, Xi'an Key Laboratory of Special Energy Materials, Northwest University, Xi'an 710069, China; 17782835382@163.com
* Correspondence: liuenzhou@nwu.edu.cn

**Abstract:** Oil–water separation using special wettability materials has received much attention due to its low energy consumption and high separation efficiency. Herein, a fluorine-free superhydrophobic cotton fabric (PDMS/STA-coated cotton fabric) was successfully prepared by a simple impregnation method using hydroxyl-capped polydimethylsiloxane (PDMS-OH), tetraethoxysilane (TEOS), and stearic acid (STA) as precursors. The investigation found that the cross-linking reactions between the hydroxyl groups of PDMS-OH and hydrolyzed TEOS enabled a strong interaction between PDMS-OH and cotton fabric. Furthermore, a suitable roughness surface of coated cotton fabric was established by introducing STA due to its long chain structure. The contact angle of this composite can reach 158.7° under optimal conditions due to its low surface energy and desired roughness. The oil/water separation efficiency of PDMS/STA-coated cotton fabric is higher than 90% even after 10 cycles of oil–water separation, and the oil flux can reach 11862.42 L m$^{-2}$ h$^{-1}$. In addition, PDMS/STA-coated cotton fabric exhibits excellent chemical stability and durability under extreme conditions such as strong acid (HCl, pH = 1~2) and alkali (NaOH, pH = 13~14), and the hydrophobicity of PDMS/STA-coated cotton fabric was decreased to 147° after 300 cycles of abrasion testing.

**Keywords:** PDMS; STA; superhydrophobic cotton fabric; oil–water separation; high separation flux





## 1. Introduction

The production processes of industries are accompanied by the releasing amount of wastewater containing oil [1,2]. Meanwhile, the frequent occurrence of oil spills in recent years has caused severe environmental issues [3,4]. In extreme cases, these oily wastewaters can threaten the ecological environment and human health if they are not treated properly and promptly [5,6]. Currently, mechanical extraction [7], chemical degradation [8], combustion [9], and physical adsorption [10] are the most prevalent methods for dealing with oil–water separation problem. Although they can alleviate the above-mentioned pollution to some extent, the high cost, secondary pollution, low separation efficiency, and poor reusability have seriously limited their large-scale applications. Recently, porous materials such as sponges, foams, and textiles are employed to separate the oil/water mixture [11–16]. However, the capacity and flux are typically low because these materials themselves do not discriminate between water and oil, resulting in unsatisfactory separation selectivity and efficiency. Therefore, it is urgent to explore an oil–water separation material with low energy cost, high efficiency, long-term stability, and environmentally friendly features.

Since the discovery of the lotus leaf effect [17], superhydrophobic materials with high water repellency abilities can achieve self-cleaning [18], anti-corrosion [19], anti-biofouling [20], anti-fogging [21], oil–water separation [22], and so on. Preparation of superhydrophobic materials with oil–water separation ability has become a research topic. There are numerous methods for fabricating superhydrophobic materials, including spraying [23–26], impregnation [27–29], sol-gel [30–32], etching [33,34], and (electro)chemical

deposition [35–37]. Typically, substances with low surface energy, such as fluorinated silanes [38], tetrafluoroethylene [39], and fluoropolymer [40–42], were used to reduce the surface energy and enhance the roughness of systems in order to obtain superhydrophobic materials. For example, Lin [43] reported a dual-functional superhydrophobic photothermal coating on glass by a chemical vapor deposition method, which was obtained by using candle soot (CS) and 1 H, 1 H, 2 H, 2 H-perfluorodecyltrimethoxysilane (PFDTMS) as raw materials, and the coating exhibited superhydrophobic and great anti-bacterial and anti-biofilm formation properties. Chen [44] fabricated a superhydrophobic 316L stainless-steel mesh by using perfluorooctanoic acid as low-energy material through an anodic oxidation method, and the separation efficiency of various oils was above 95%. However, these fluorine-containing materials are usually costly, complex fabrication processes and the existence of secondary contamination [45]. Therefore, the exploration of fluorine-free materials had received much attention. Tagliaro [46] reported a sustainable fluorine-free transparent coating, which was obtained by modification with fatty acid side groups and then deposited through a solvent-free deposition method, showing good durability with high hydrophobicity. Xu [47] reported a versatile fluorine-free method for fabricating superhydrophobic materials by adding a C9 petroleum resin blend to a styrene–butadiene–styrene (SBS) solution and then using a hydrophobic $SiO_2$ nanoparticle solution to obtain superhydrophobic coatings dispersion, which can create superhydrophobic surface through spraying it on different substrate surfaces and exhibit great water repellency. However, like with most superhydrophobic material preparation methods, the aforementioned cases frequently involve time-consuming, expensive, and specialized equipment, which restricts their practical application [48–52]. Cotton fabric, on the other hand, is a low-cost, high-yield natural fiber that is popular in many fields due to its comfort, softness, breathability, wash resistance, and moisture absorption qualities [53–58]. Therefore, it is crucial to combine their advantages.

Herein, a robust fluorine-free superhydrophobic cotton fabric is obtained by a simple impregnation process using polydimethylsiloxane (PDMS-OH) and stearic acid (STA) as precursors. The hierarchical surface structure of cotton fabric not only increases the fabric's roughness but also captures air to form an air cushion between the coating and the water. In addition, the cross-linking reaction between the hydroxyl groups of PDMS-OH and tetraethoxysilane (TEOS) during the condensation reaction can further strengthen the connection between the superhydrophobic coating and the cotton fabric. The obtained superhydrophobic cotton fabric has excellent resistance durability, chemical stability, self-cleaning performance, and a high oil–water separation efficiency with a desired separation flux.

## 2. Materials and Methods

### 2.1. Materials

Hydroxyl-terminated polydimethylsiloxane (PDMS-OH, cst = 40) was purchased from Macklin Reagent Co., Ltd. (Shanghai, China). Tetraethoxysilane (TEOS, AR), sodium hydroxide (NaOH, 96%), and hydrogen chloride (HCl, 96%) were obtained from Damao Chemical Reagent Factory. Bis(lauroyloxy)dioctyltin (DOTDL, 96%) was received from Shanghai Yien Chemical Technology Co., Ltd (Shanghai, China). Stearic acid (STA, CP), n-hexane (AR), and anhydrous ethanol (AR) were provided by Tianjin Fuyu Fine Chemical Co., Ltd. (Tianjin, China) Methylene blue (MB) and rhodamine B (Rh B) were purchased from Beijing Chemical Works. Sandpaper (360 meshes), peanut oil, and cotton fabric (CF) were bought from a local market. All chemicals were used as received without further purification, and ultra-pure water was used during all experiments. The pristine CF was ultrasonically cleaned with anhydrous ethanol and water in order to remove the surface impurities and then dried in an oven for further procedures.

### 2.2. Fabrication of Superhydrophobic Cotton Fabric

Figure 1 shows the fabrication processes of superhydrophobic cotton fabric. Specifically, 1.0 g PDMS-OH and 4.0 g TEOS were added into 30 mL n-hexane solution for

cross-linking with the assistance of 0.1 g DOTDL catalyst at room temperature for 4 h. Subsequently, 8 wt% STA was heated at 75 °C in 40 mL n-hexane until it was dissolved completely. Then, the pretreated CF (60 mm × 60 mm) was immersed in the above mixture for 24 h. Finally, the coated CF was dried at 60 °C to obtain superhydrophobic PDMS/STA-coated cotton fabric (PS-CF). Furthermore, PDMS-OH and STA were used respectively to treat CF under the same condition, and the products were marked as P-CF and S-CF.

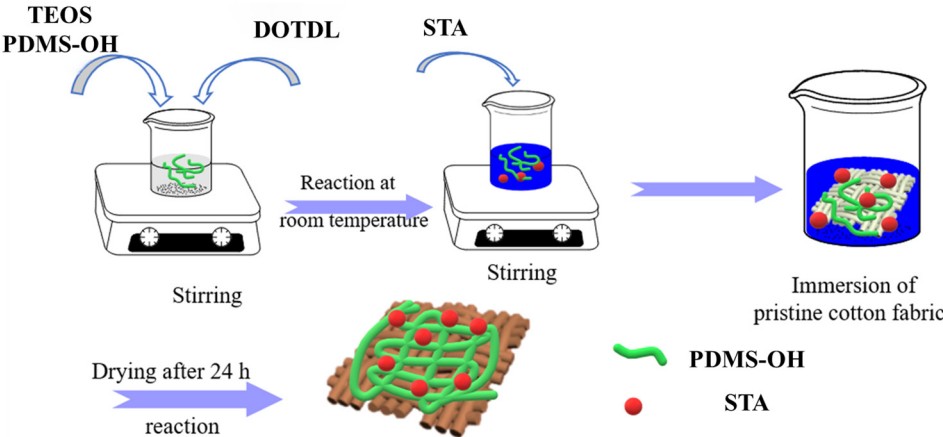

**Figure 1.** Preparation processes of PDMS/STA-coated CF.

### 2.3. Oil–Water Separation

Figure 2 shows the schematic illustration of the oil/water separation device and process. The oil–water mixture contains 50 mL of water and 50 mL of different oil, including dichloromethane, trichloromethane cyclohexane, peanut oil, and n-hexane. The oil and water were pre-stained with rhodamine B (Rh B) and methylene blue (MB) for clear observation, respectively. The oil–water separation experiment was conducted by laying the prepared PS-CF at the mouth of the oil–water separation device (25 mm × 25 mm). The oil–water mixture is poured into the above device, and the separation process is conducted with the help of gravity. Additionally, the time for the whole separation process and the amount of separated oil are recorded. After each cycle, the PS-CF was washed with anhydrous ethanol and deionized water and then dried at 60 °C before the next separation cycle.

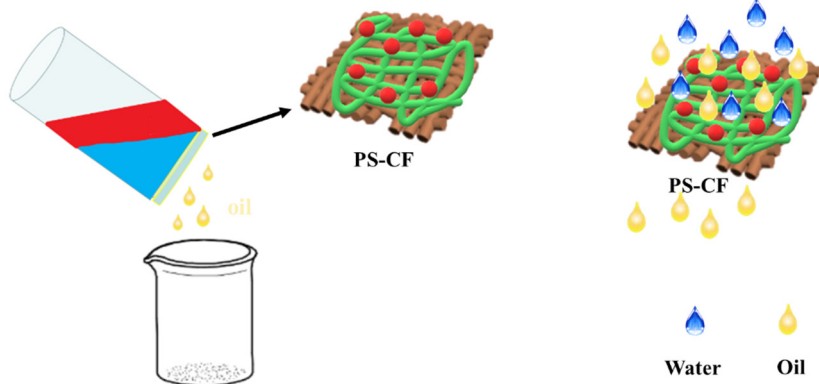

**Figure 2.** Schematic illustration of oil/water separation device and process.

The oil–water separation efficiency (*Es*) is defined as follows:

$$E_S = \frac{V_a}{V_b} \times 100\%$$

where $V_a$ represents the volume of the oil after the separation process and $V_b$ represents the volume of the oil before the separation process.

The formula for calculating the permeate flux (*J*) during the oil–water separation of various types of oils is as follows:

$$J = \frac{V}{S \times T}$$

where $V$(L) represents the total volume of oil penetrated, $S$(m$^2$) represents the effective penetration area during oil–water separation, and $T$(h) represents the duration of the oil–water separation process.

### 2.4. Characterizations

The water contact angle (WCA) was calculated using a Contact Angle System OCA20 (Dataphysics, Germany) with 2~3 μL deionized water at room temperature. The WCA was the average of three different positions of the samples. X-ray photoelectron spectroscopy (XPS, Kratos AXIS NOVA spectrometer) was used. All binding energies were calibrated using the C 1 s peak at 284.8 eV as the reference. Fourier transform infrared spectra (FT-IR, PerkinElmer Frontier, Waltham, MA, USA) were employed to reveal the chemical composition of the samples. The morphologies of samples were observed by the scanning electron microscope (SEM, Carl Zeiss SIGMA) coupled with X-ray energy-dispersive spectrometer (EDS). Ultrasonic cleaning machine (KQ600-KDE, 600 W, Kunshan ultrasonic instruments Co., Ltd., Kunshan, China) was used at 600 W.

## 3. Results and Discussion

### 3.1. Wettability of Superhydrophobic Cotton Fabric

Figure 3a,b shows the states of various liquids (water, tea, orange juice, peanut oil, milk, N-hexane, and cyclohexane) on the surface of superhydrophobic PDMS/STA-coated cotton fabric and original cotton fabric. It can be observed that the water-based liquid droplets on the composite are spherical, including milk, tea, and juice, and the oils such as n-hexane and dichloromethane are in a completely wetted state. As shown in Figure 3c, when the tilt angle of the fabric is less than 10°, the water droplets can slide down the PS-CF surface quickly, indicating that the cotton fabric prepared in this work has superhydrophobic properties. It is known from the Cassie–Baxter equation that this non-wetting phenomenon results from the low adhesion caused by the air cushion, which makes the water droplets easily roll off the surface of the PS-CF. Then, the PS-CF was immersed in the water (Figure 3d), and the phenomenon of "silver mirror" can be observed over the surface of PS-CF, which results from a large number of air pockets created by the micro- and nanostructure on the surface of PS-CF, suggesting that the above superhydrophobic phenomenon belongs to Cassie–Baxter state [59].

### 3.2. Formation of Superhydrophobic PDMS/STA-Coated Cotton Fabric

Due to the presence of a large number of hydroxyl groups on cellulose, the primary component of cotton fabric, the original cotton fabric is hydrophilic with WCA lower than 90°. In this work, PDMS-OH and STA were used to reduce the surface energy of cotton fabric with an appropriate roughness. The silane-coupling agent TEOS was employed to cross-link with PDMS-OH to form an irregular mesh structure on the surface of cotton fabric under the assistance of DOTDL, and hydrolyzed TEOS, PDMS-OH, and cotton fabric can form strong connections by the condensation reaction.

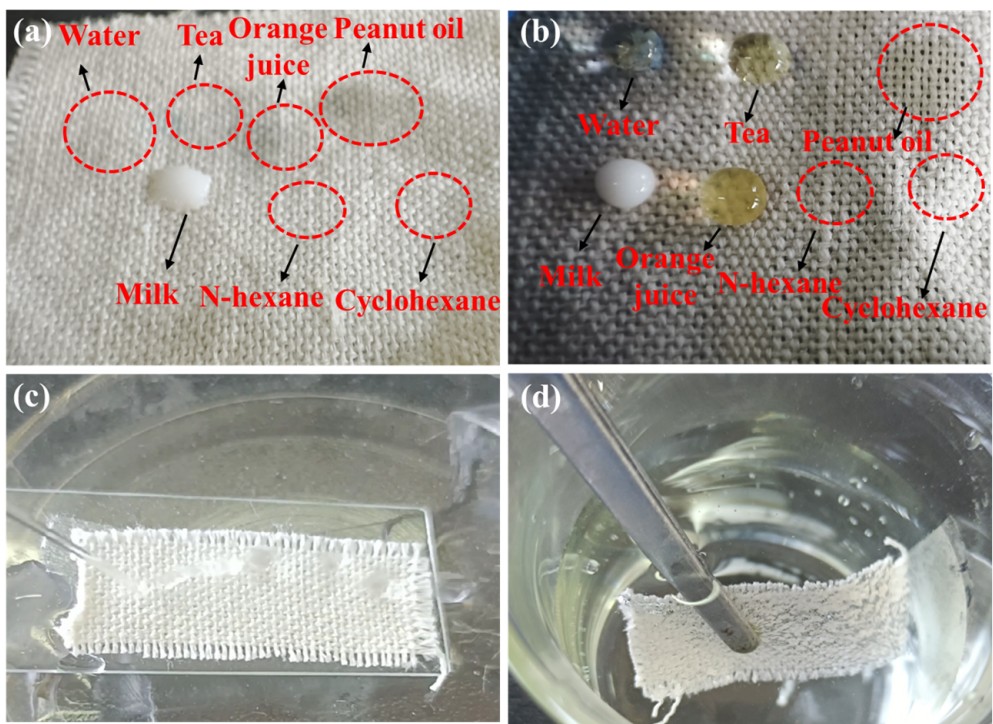

**Figure 3.** Images of (**a**) liquid droplets on the CF, (**b**) liquid droplets on PS-CF, (**c**) water rolling off from the surface of PS-CF, and (**d**) PS-CF in water.

FT-IR spectroscopy was used to determine the chemical compositions of untreated cotton fabric and PDMS/STA-coated superhydrophobic cotton fabric. As shown in Figure 4a, the peak at 3334 cm$^{-1}$ from untreated cotton fabric corresponds to the stretching vibrations of –OH, and peaks at 2914 and 2851 cm$^{-1}$ stem from –CH$_2$ symmetrical stretching vibrations. Moreover, a strong peak at 1026 cm$^{-1}$ is associated with –OH bending vibration and C–O–C stretching vibration. For the PDMS/STA-coated superhydrophobic cotton fabric, a new peak at 2964 cm$^{-1}$ corresponds to the stretching vibration of the –CH$_3$ group from PDMS-OH, indicating PDMS is successfully deposited on the surface of cotton fabric. The signal at 1260 cm$^{-1}$ belongs to the –CH$_3$ stretching vibration from Si–CH$_3$, further indicating the presence of PDMS-OH elastomer [60]. In addition, a new peak at 794 cm$^{-1}$ originates from the Si–O–Si symmetric stretching vibration in the composite. The peak at 1697 cm$^{-1}$ is associated with C=O stretching, confirming the successful implementation of STA. Moreover, the peak intensity at 3334 cm$^{-1}$ is clearly reduced, implying the surface –OH groups are consumed by the cross-linking reaction with PDMS-OH and hydrolyzed TEOS [61]. Furthermore, EDS was utilized to analyze the surface chemical composition of the composite. C (51.94 wt%), O (40.16 wt%), and Si (7.9 wt%) were evenly distributed on the surface of the PDMS/STA-coated cotton fabric in Figure 4b. The above results indicate that the cotton fabric coated with PDMS and STA was successfully fabricated.

Figure 5 presents the XPS spectra of the samples. The survey spectrum implies that the pristine cotton fabric is composed of elements C and O in Figure 5a. After introducing PDMS-OH, TEOS, and STA, a new Si signal belonging to Si 2$s$ and Si 2$p$ is detected at 153.08 eV and 102.08 eV in Figure 5b. Moreover, the content of the O element on the surface of PS-CF increases substantially and the content of the C element decreases. This is due to the large amount of -COOH in STA, which leads to a corresponding increase in the O element. This result further proves that the PS-CF surface is covered by the cross-linked network formed by PDMS and STA. Furthermore, the high-resolution XPS C1$s$ and Si 2$p$ spectra of the samples were studied. The C1$s$ of CF can be curve-fitted into typical peaks of C-O, C=O, and C-C with binding energies at 288.7, 286.1, and 284.8 eV, respectively. As shown in Figure 5c,d, the intensity of C=O and C-C groups in the composite increases after

introducing PDMS/STA coating, which is consistent with FT-IR analysis. Comparing the Si 2*p* high-resolution profiles of the CF and PS-CF in Figure 5e,f, it can be seen that the orbital peaks located at 102.28 eV and 103.78 eV are attributed to the Si-C and Si-O bonds, respectively. The above results indicate the successful loading of PDMS and STA on the surface of cotton fabrics.

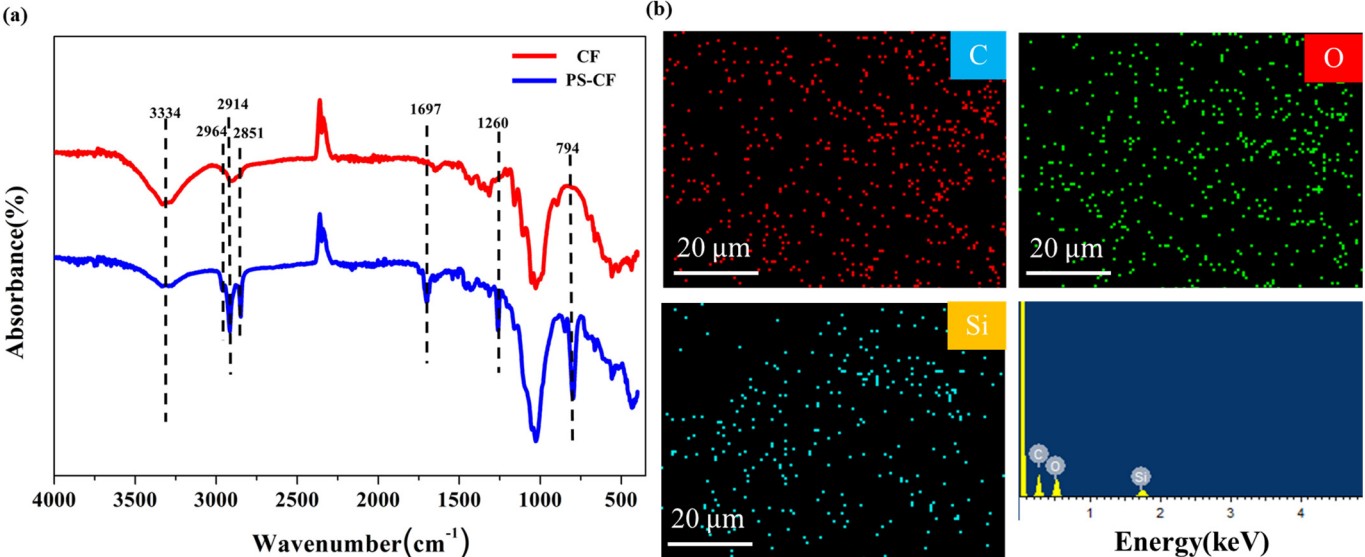

**Figure 4.** (**a**) FT-IR spectra of CF and PS-CF. (**b**) Element distribution map and curve of PS-CF.

In order to further investigate the relationship between the surface micromorphology and material wettability of CF, PCF, SCF, and PS-CF, the surface micromorphology of samples was observed by SEM. Figure 6a shows the SEM image of the original cotton fabric, and it is relatively smooth overall with fine protrusions distributed on the cotton fabric fibers. Figure 6b is the SEM image of PCF, which was treated with PDMS-OH only; compared with the original cotton fabric, its surface is obviously smoother and flatter, a layer of film-like material is uniformly attached to the surface of the fabric fibers, and the texture of the original cotton fabric fibers is reduced. This is due to the uniform film formed by the cross-linking reaction between PDMS-OH and TEOS on the surface of the cotton fabric, which reduces the surface tension of the fabric [62]. Figure 6c shows the surface of the SCF sample treated with STA only, and the fabric surface is obviously rough; compared to CF and PCF, STA is unevenly distributed on the fabric surface, and the fabric fiber protrusions increase and some fibers are broken. This is because STA is simply attached to the surface of cotton fabric and not firmly. As shown in Figure 6d, the protrusions on the fabric fiber surface are flat and protrude significantly more, and STA on the fabric surface shows flower morphology and is more uniformly distributed on PS-CF samples compared to SCF, and the overall roughness of the fabric increases significantly. The WCAs of PCF, SCF, and PS-CF are 132°, 138°, and 154°, respectively. It shows that the synergistic effect of PDMS and STA satisfies the low surface energy and suitable roughness required for building superhydrophobic surface materials.

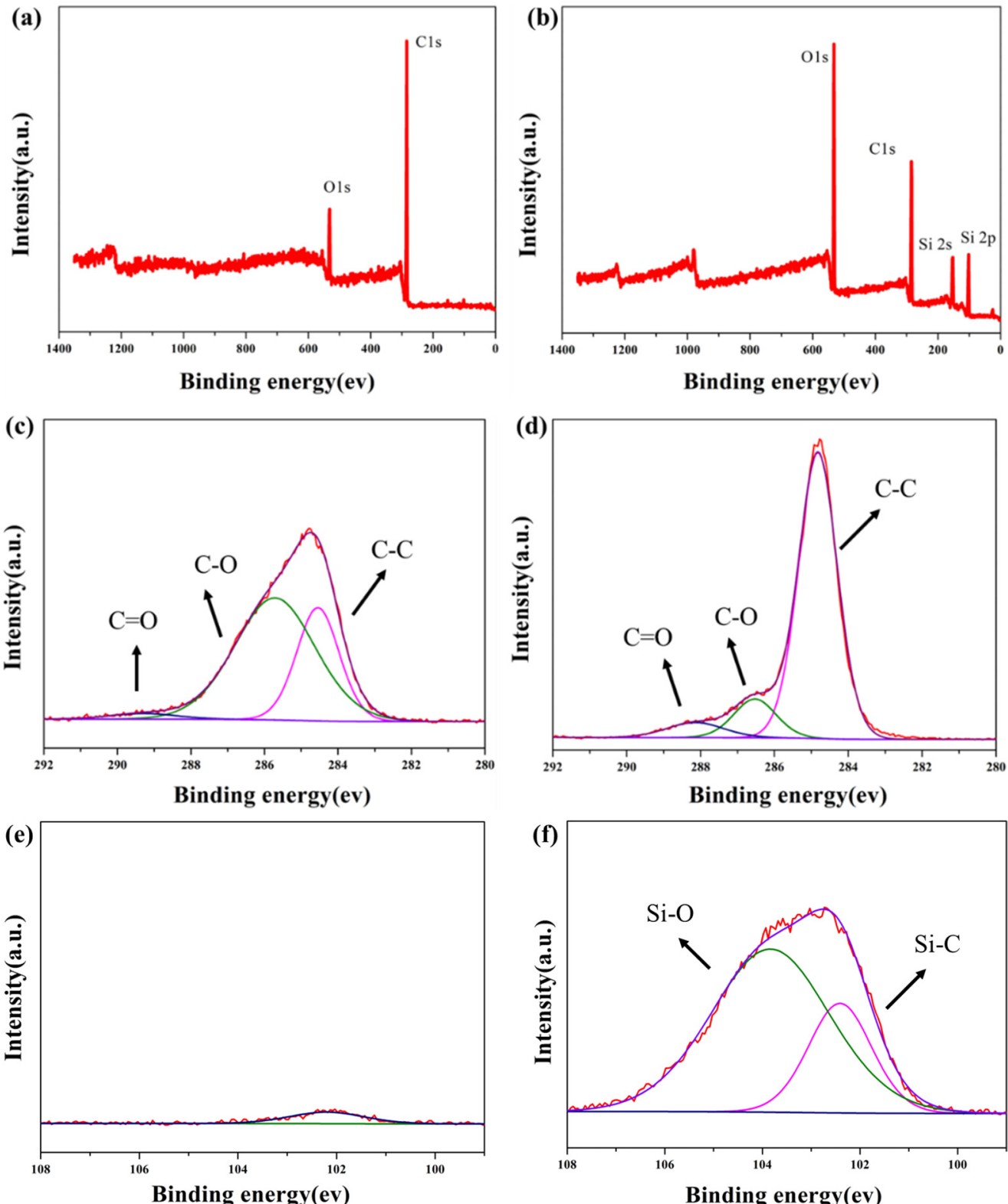

**Figure 5.** (**a**,**b**) XPS survey spectra, (**c**,**d**) high-resolution spectra of C1*s*, and (**e**,**f**) high-resolution spectra of Si 2*p* of CF and PS-CF sample.

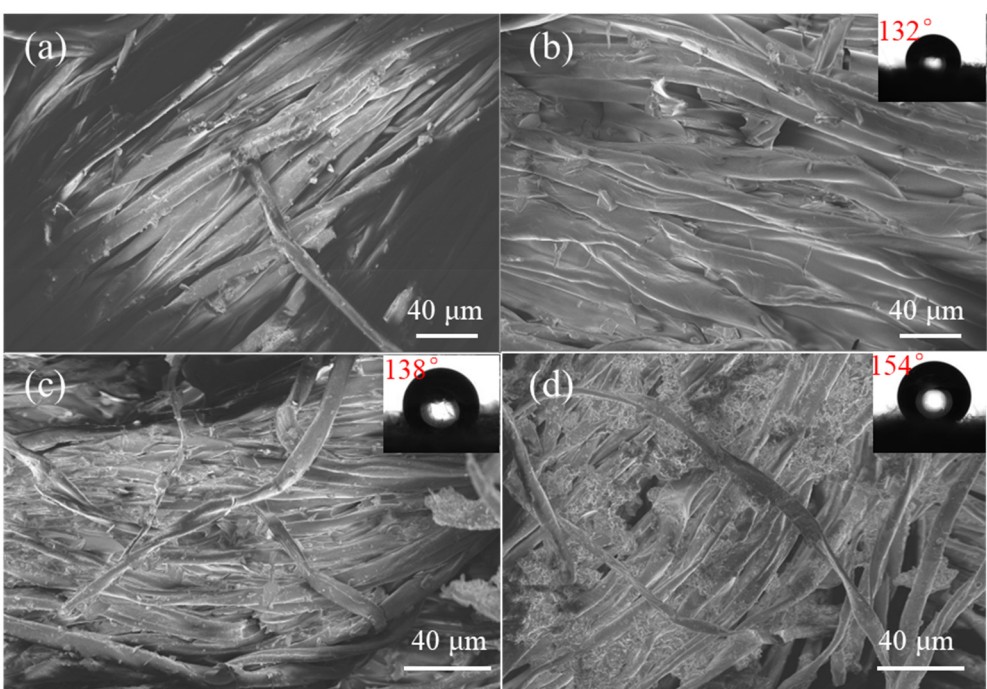

**Figure 6.** SEM images of (**a**) CF, (**b**) PCF, (**c**) SCF, and (**d**) PS-CF.

*3.3. Oil–Water Separation Performance*

The PS-CF has superhydrophobic and lipophilic properties, which enable it to be used in the field of oil–water separation. When the oil–water mixture passes through the oil–water separation device with PS-CF as the filter membrane under the action of gravity, the water phase is trapped above the fabric, while the oil phase can wet the PS-CF and pass through the membrane, thus realizing the separation of oil and water phases. Oil–water separation experimental results are shown in Figure 7a. The separation efficiency of the dichloromethane/water mixing system is 98.2%, the trichloromethane/water system separation efficiency is 97%, n-hexane/water system and cyclohexane/water system results are closer, that is, 93.2% and 93.8%, respectively, and the separation efficiency of the peanut oil/water mixed system was only 90.4%. On the one hand, dichloromethane and trichloromethane, as representatives of heavy oils, are denser than water, and dichloromethane and trichloromethane are more likely to penetrate through the aqueous-phase layer and thus through the PS-CF filter layer under gravity than peanut oil, n-hexane, and cyclohexane, which are lighter in density than water. On the other hand, considering the difference in viscosity and physicochemical properties of the oils, peanut oil is more viscous than the other four oils, which is more likely to be adsorbed in the pores of the substrate when passing through the cotton fabric as the porous substrate material, making the pores of the substrate smaller or even blocked, resulting in lower separation efficiency and permeation flux of different oil.

The oil flux is another significant indicator for evaluating the oil–water separation process. Therefore, five different oil fluxes on PS-CF were measured. Heavy oil and light oil had different separation fluxes due to their different density, as shown in Figure 7b. The permeate fluxes of dichloromethane and trichloroformethane can reach 11,862.42 $L \cdot m^{-2} \cdot h^{-1}$ and 9320.71 $L \cdot m^{-2} \cdot h^{-1}$, respectively, and the permeate fluxes of peanut oil, n-hexane, and cyclohexane in water were 4926.67, 5921.55, and 6680.73 $L \cdot m^{-2} \cdot h^{-1}$, respectively. Zhang [62] fabricated a superhydrophobic cotton fabric using beeswax/lignin compound through the spraying method, and it can absorb oil from an oil/water mixture. The recovery rate of modified cotton for trichloromethane/water separation reaches at least 88.92% and the highest is 93.78%. Wen [63] used the graft polymerization method to graft polymer brushes on the cotton surface for the immobilization of Ag, which was subsequently coated with PDMS; the coated cotton fabric can separate oil from pure water and artificial seawater,

and the separation efficiency is 97%. Li [64] used fluorosurfactant to modify hydrophilic $Al_2O_3$ nanoparticles by mixing it in the ethanol solution, and then cotton fibers were immersed in the above solution to get coated. As a result, the efficiency is 98% higher than both diesel-in-water and hexadecane-in-water separation with 520 $L·m^{-2}·h^{-1}$ to 650 $L·m^{-2}·h^{-1}$ oil permeate flux. Prasanthi I reports a PDMS sponge, which was prepared by the sugar-cube templating approach using fluorinated graphite polymer, and PDMS sponges can separate the oil–water emulsion by adsorbing oil from oil–water emulsions with 90–99% separation efficiencies. Meanwhile, Qiu [65] obtained a PDMS sponge via an ultrasound-assisted in situ polymerization approach, which can absorb 3 to 19 times its own weight oil with approximately 89% separation adsorption rate on n-hexane [66]. Therefore, PS-CF has relatively high separation efficiency and oil permeate flux.

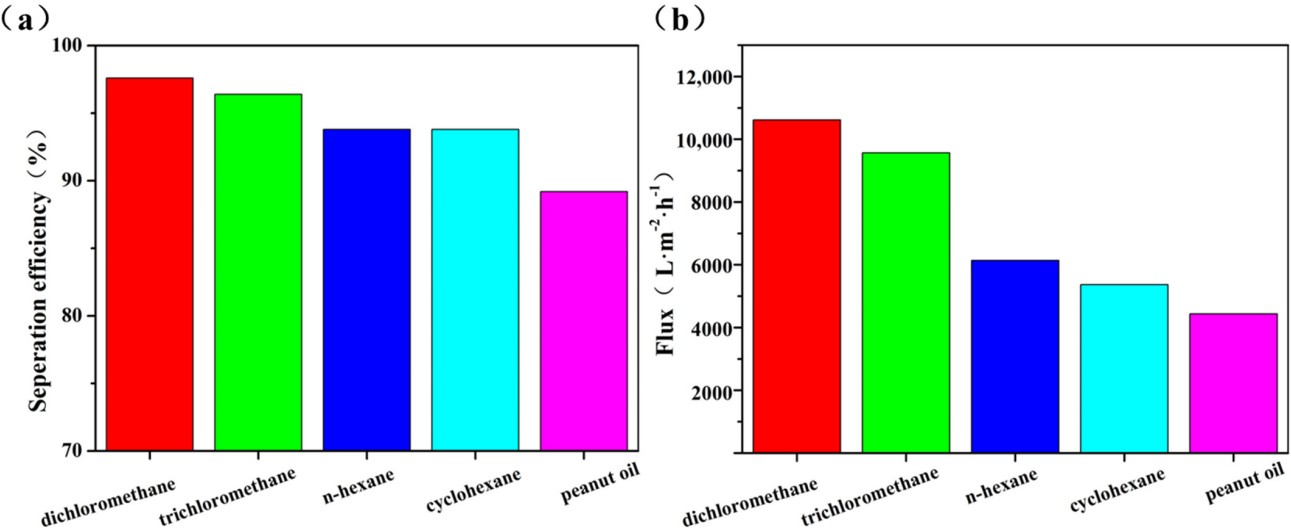

**Figure 7.** (**a**) The separation efficiencies and (**b**) the oil flux during oil–water separation.

In addition, PS-CF performs well in the oil–water separation cycle experiment. The dichloromethane/water system was chosen to conduct the cycle experiment, and the results are depicted in Figure 8. Although the overall oil–water separation efficiency decreases with increasing cycle times, the oil–water separation efficiency of the superhydrophobic cotton fabric can be maintained at 92.8% after 10 cycles.

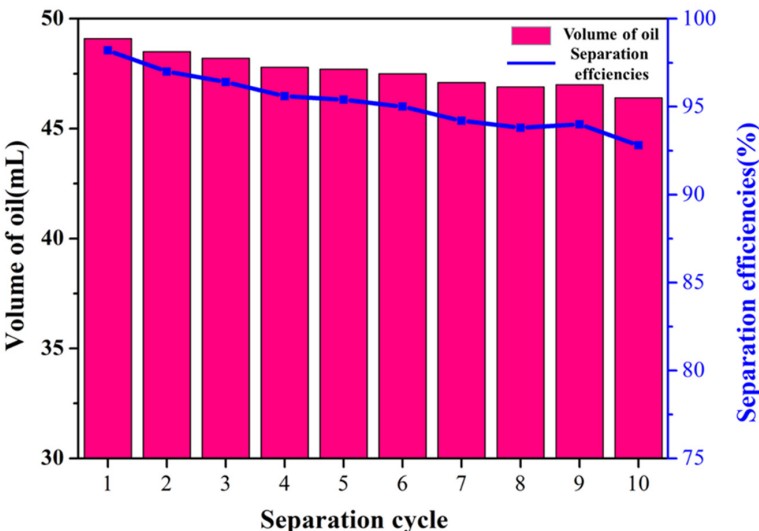

**Figure 8.** Association between separation efficiency and the number of cycles of dichloromethane/water.

### 3.4. Durability and Chemical Stability of PS-CF

Superhydrophobic fabrics are unavoidably required for applications in harsh environments; therefore, superhydrophobic materials should withstand harsh conditions. Several sets of experiments have been designed in order to verify the usability of PS-CF in extreme conditions. As shown in Figure 9a,b, to investigate the mechanical stability, we laid the sandpaper flat on the experiment table, attached the PS-CF to one side of the slide with double-sided tape, put a 150 g weight on the other side of the slide, and then moved the slide in the horizontal direction by traction for 10 cm. This process was regarded as one abrasion cycle. Until 300 cycles of abrasion, the PS-CF remains superhydrophobic with WCA higher than 150° in Figure 9c, whereas after 300 cycles of abrasion and beyond, the as-cotton fabric still maintains a good water repellency. Figure 9c displays the WCA after the abrasion test, inset is the SEM of fabric after 400 abrasion cycles, and the WCA of PS-CF is 146.4°, indicating that PS-CF still has good mechanical strength due to the cross-linking network structure and the strong binding force between PDMS, STA, and chemical bonds formed on the surface of cotton fabric.

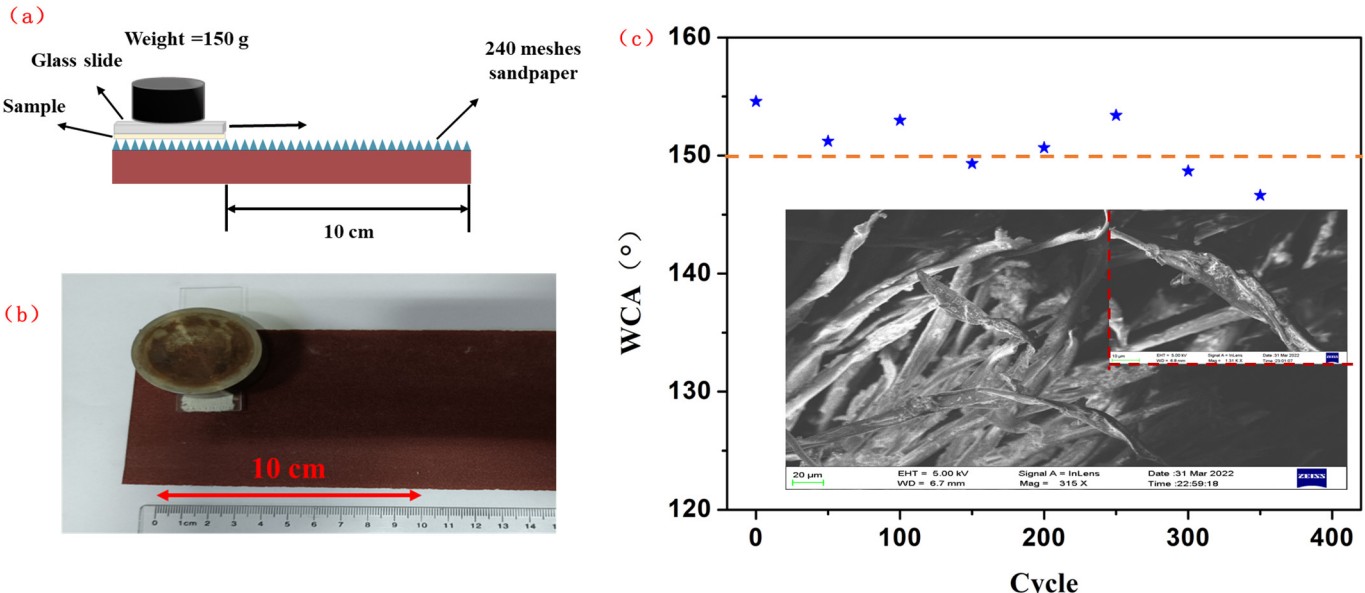

**Figure 9.** (**a**,**b**) Schematic of mechanical durability tests. (**c**) WCA of different abrasion cycles (inset is SEM image after 400 abrasion cycles).

As a material for oil–water separation applications, PS-CF will inevitably face scenarios where it is applied in extreme environments. Therefore, PS-CF should have good physicochemical stability, and the chemical stability of PS-CF was tested. After the PS-CF fabric was completely immersed in deionized water, strong alkali solution (NaOH, pH = 13~14), ethanol solution, and strong acid solution (HCl, pH = 1~2) for a certain time, the WCA of the sample was measured after washing and drying with deionized water. Figure 10 shows the graphs of the changes of WCA on the surface of PS-CF after immersion in strong acid, strong base, ethanol, and deionized water for 12, 24, 36, 48, and 72 h. The WCA of PS-CF decreases to 150.7° and 150.2° after immersion in water and NaOH solutions for more than 72 h. It still has the superhydrophobic property. In contrast, the WCA of PS-CF after immersion in ethanol and HCl solutions gradually decreases, and after 36 h immersion, the WCA of the samples is lower than 150°, and the WCAs of PS-CF samples are 142.2° and 140.5° after 72 h immersion; although it loses the superhydrophobicity it keeps good hydrophobicity.

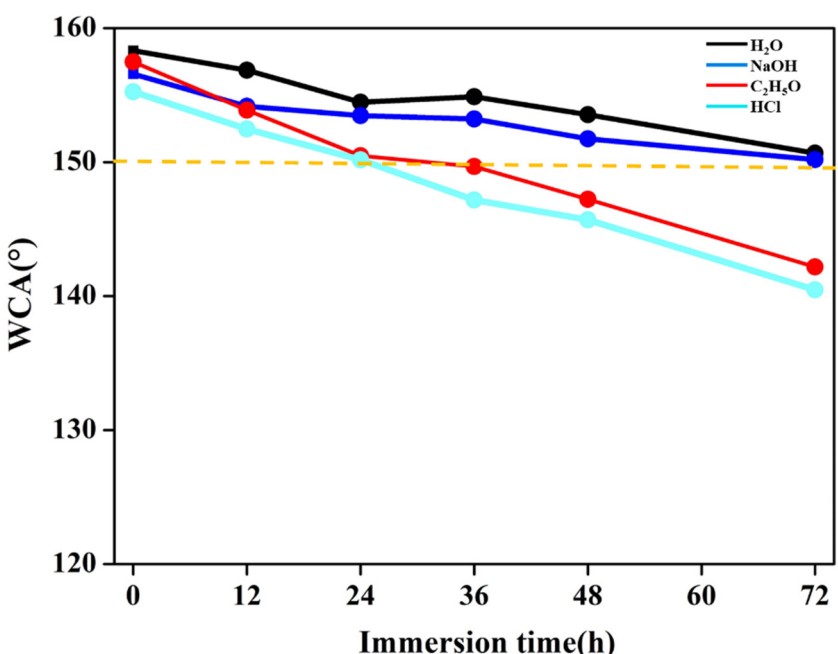

**Figure 10.** WCA change with time after PS-CF immersion in different solutions.

In order to further investigate the effect of different solutions on PS-CF, SEM characterization was performed on the samples after 72 h immersion. Figure 11a–d are the SEM images of the surface of PS-CF after immersion in deionized water, NaOH, ethanol, and HCl for 72 h. As shown in Figure 11a,b, the fabric fiber surface of the PS-CF samples did not change significantly after 72 h of immersion in deionized water and NaOH, and there was no obvious trace of damage to its micro- and nanostructure, so the samples could still maintain their superhydrophobic state after immersion. Whereas the fabric fiber surface of the PS-CF samples shown in Figure 11c,d, after immersion in ethanol and HCl solutions for 72 h, obviously had less adhesion, and the surface was smoother and flatter; the surface rough structure was damaged. This is because the cross-linked network formed by the condensation of silanol bonds and hydroxyl groups of PDMS-OH on the fabric surface was destroyed by the strong acidic HCl, resulting in the loss of strong bonding between the particles on the surface of the cotton fabric; as a result, some of the STA particles fell off, and hydrophobic properties of the samples were reduced. With a long time immersion in the ethanol solution, some PMDS-OH and STA will be shed due to dissolution. However, in the ambient environment, the solubility of PDMS-OH and STA in ethanol solution is relatively low; consequently, the WCA on the surface of SPCF after immersion in ethanol solution for the same time is slightly higher than in HCl.

The low-surface-energy substances used in the construction of superhydrophobic surface materials and the particulate materials or structures that provide roughness can be peeled off or destroyed during application due to vibration or other factors, resulting in a decrease in the hydrophobic properties of the materials. To estimate the peeling resistance of the material, ultrasonic peeling resistance testing was conducted on PS-CF.

Figure 12 shows the results of sonicating PS-CF adhered to a slide glass with double-sided adhesive and placed in an ultrasonic cleaner at 600 W. After 12 h of sonication peeling, the SPCF had a WCA of 152.2° and still maintained superhydrophobic. This is because the PDMS-OH and STA on the surface of the fabric are chemically bonded, instead of being simply mechanically attached. Once again, the test results indicate that PS-CF has excellent durability.

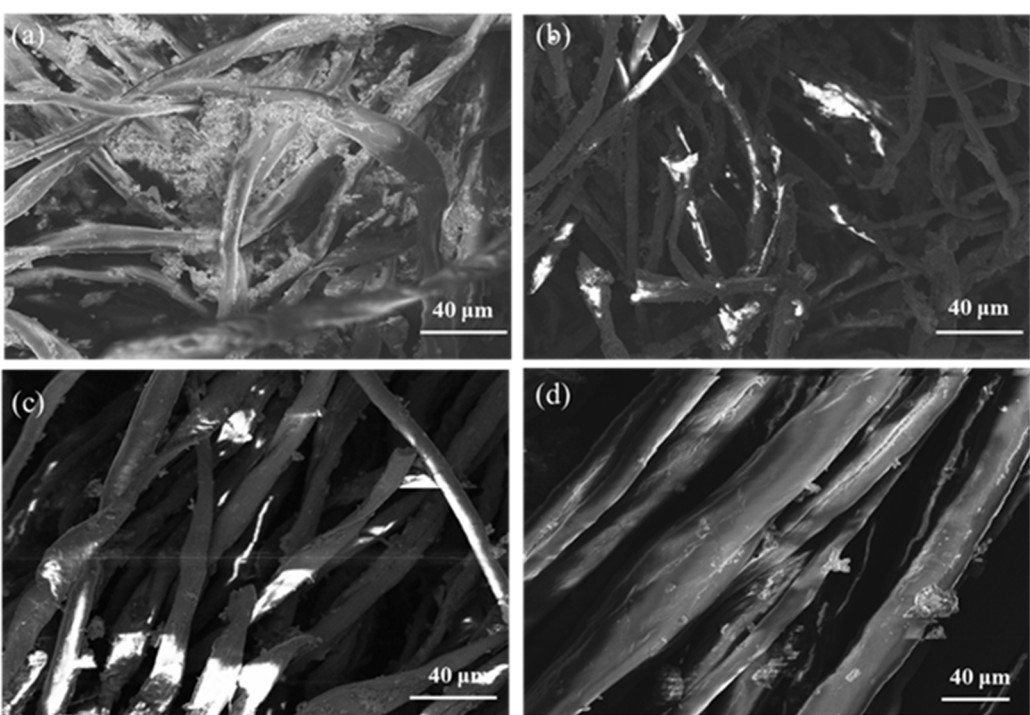

**Figure 11.** SEM images of PS-CF after 72 h immersion in (**a**) $H_2O$, (**b**) NaOH, (**c**) $C_2H_5OH$, and (**d**) HCl.

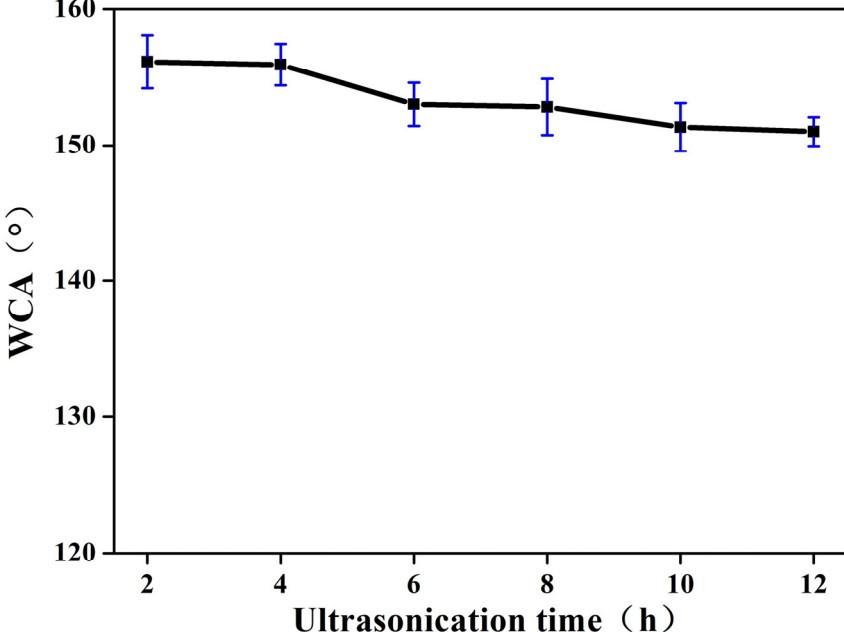

**Figure 12.** The relationship between WAC and ultrasonication time.

## 4. Conclusions

In summary, a superhydrophobic cotton fabric with great durability, fluorine-free, and low cost was fabricated through the impregnation method. Due to the low-surface-energy material and hierarchical rough surface, the surface static water contact angle (WCA) of the superhydrophobic cotton fabric can reach 158.6°. Moreover, as-prepared PS-CF can maintain superhydrophobicity in alkaline environments for over 72 h, withstanding over 300 abrasion cycles. Additionally, the oil–water separation efficiency of PS-CF is higher

than 90% when it is applied to various oil–water mixed systems and maintains a better separation efficiency with the high-level flux of oil.

**Author Contributions:** Conceptualization, D.T.; methodology, D.T. and E.L.; validation, D.T.; formal analysis, D.T.; investigation, D.T.; resources, D.T.; data curation, D.T.; writing—original draft preparation, D.T.; writing—review and editing, D.T. and E.L.; visualization, D.T. and E.L.; supervision, E.L.; project administration, E.L.; funding acquisition, E.L. All authors have read and agreed to the published version of the manuscript.

**Funding:** This work was supported by the Natural Science Basic Research Program of Shaanxi Province: 2023-JC-YB-115; Key Science and Technology Innovation Team of Shaanxi Province: 2022TD-33; National Natural Science Foundation of China: 11974276; National Natural Science Foundation of China: 22078261.

**Institutional Review Board Statement:** Not applicable.

**Informed Consent Statement:** Not applicable.

**Data Availability Statement:** Data are contained within the article.

**Conflicts of Interest:** The authors declare no conflict of interest.

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
