# Peer review of "Facile Fabrication of Robust and Fluorine-Free Superhydrophobic PDMS/STA-Coated Cotton Fabric for Highly Efficient Oil-Water Separation"

_coatings, doi:10.3390/coatings13050954_

Round 1
Reviewer 1 Report
In " Facile Fabrication of Robust and Fluorine-Free Superhydrophobic PDMS/STA-Coated Cotton Fabric for Highly Efficient Oil-Water Separation," by Tang et al., the authors report the fabrication of a fluorine-free superhydrophobic cotton fabric for oil-water separation. The fabric was prepared by impregnating hydroxyl-capped polydimethylsiloxane (PDMS-OH), tetraethoxysilane (TEOS), and stearic acid (STA) onto the cotton fabric. The roughness of the coated fabric was achieved through the introduction of STA, which also led to a high contact angle of 158.7°. The proposed fabric exhibited high oil/water separation efficiency, with an efficiency of over 90% even after ten cycles of separation. Finally, the PDMS/STA-coated cotton fabric demonstrated excellent chemical stability and durability under extreme conditions. In my opinion, this paper is well organized and written. However, there are some issues that are not fully addressed and it could be further improved by addressing the following comments and suggestions.
1. Is this fabrication method available for large-scale and mass production in the textile industry? What is the expected cost (e.g., materials and chemicals except for labor) for the fabrication of the proposed fabric?
2. As life goods, the materials used in this work should be proven to be biocompatible, especially for skin-related applications (i.e., skin allergy).
3. Please check the language of the legend in Figures 2a and 2b, as they appear to be in Chinese. Additionally, kindly ensure that the liquid type used in the figure is clearly specified.
4. Please adjust the y-axis range in Figure 3a to ensure that the variations and peak positions are clearly visible.
5. The legend in Figure 9a also appears to be in Chinese.
6. For applications in harsh environments, I would recommend not only conducting a laundry test with at least three repetitions using laundry detergent, but also evaluating the thermal stability of the proposed fabric.
7. Can the authors provide the dynamic water contact angles of the proposed fabric?
Overall, fine/minor spell check in the English language and style should be required.
Reviewer 2 Report
The paper describes an interesting impregnation strategy to obtain robust fluorine-free cotton fabric coating for highly efficient oil-water separation. The coating exhibits good adhesion, resistance to mechanical action and durability in high alkaline and acid conditions
I have a few remarks that the authors should address:
- - Probably on line 99 and in Fig. 1 the authors wanted to write PDMS- OH instead of PDMS
- - The concentration of stearic acid solution should be specified on line 101
- - Please clarify what the authors mean for “completely saturated state” on line 122
- - Chinese words should be converted to English words in Fig. 2 and 9
- - Probably the authors intended C=O and C-C (instead of C=O and C-O) on line 173. Morover the condensation reactions of the cellulose C-OH with the added molecules shouldn’t give Si-O-C instead of the Si-C indicated on line 176 and Fig. 4d? Please clarify
- - Shoudn’t Es be related to Vb/Va instead of the reverse so as indicated on line 219?
- - What does represent the red band in Fig.6? In the experiment described in Fig.6 is the oil/water system in the emulsion state?
- The authors state on line 285-286 that : “after 350 cycles of abrasion the PS-CF remains …..”.This is not consistent with Fig. 9c showing that the limit is 300 cycles. Please clarify
- - Details about sonication experiment should be given, in particular the energy of the apparatus
- For the convenience of the reader I suggest to move the description of the O/W separation device, mechanical durability test and chemical stability test to the experimental section.
I find the english quite good
Reviewer 3 Report
The current version of the manuscript has lots of text, schemes resemblance with Progress in Organic Coatings Volume 132, July 2019, Pages 100-107 and Nanoscale, 2022,14, 5840-5850. Obviously this is not good and authors needs to highlight and improvise the text and rewrite to make the current version better. Too many figures in the manuscript which needs regrouping/rearranging and additional images to improvise. The manuscript is not brief, explanation for mechanisms, comparisons with literature, grammar usage is poor. Statements are not complete. Additional comments are given below. The current version needs a drastic changeover and proper justification with modified PDMS reports (including fluorinated derivatives).
The images needs a through changeover. The present ones look very bad.
eV should be therein fig 4.
What about the O1s high resolution XPS?
Comment on the necessity of separating peanut oil as compared to other organic solvents. Engine oil, crude oil or diesel can be separated.
Authors can compare the organics/oil sorption capacity and efficiency with state of art graphene-like PDMS sponges Eg. RSC Adv., 2017, 7, 10479, Mater. Chem. Front., 2021,5, 6244, ACS Appl. Nano Mater. 2022, 5, 5857, Chemosphere, 2021, 271, 129827.
The separation and oil sorption data needs to be stressed comparison with respect to other works.
Fig 9 a?? needs a changeover. It is not understandable to all section of audience.
The resolution of the WCA needs to be mentioned wrt instrument.
WCA change variation wrt base and acids needs to be elaborated and compared with other state of art PDMS works.
Definitely this paper doesn’t deserve to have 12 images. I suggest 6 proper images needs to be arranged with proper justification and appropriate comparison with PDMS articles.
Needs through correction
Reviewer 4 Report
The paper entitled “Facile Fabrication of Robust and Fluorine-Free Superhydropho-bic PDMS/STA-Coated Cotton Fabric for Highly Efficient Oil-Water Separation” focus on oil-water separation using special wettability materials. Efficient separation of oil and water, including emulsified oil, is one of the problems limiting the environmentally friendly development of the petrochemical industry.
Comments:
1. Figure 3. What does the signal in the 2200 cm-1 area mean?
2. What does the absorbance (%) in the FTIR spectra signify? Most of the time these values doesn't mean anything. Should the y-axis values be omitted?
3. What volume of substances (oil-water) can be separated by PDMS/STA-coated CF?
4. Authors should report thermogravimetric analysis results for PDMS/STA-coated CF.
5. The chemical stability was investigated under extreme conditions in NaOH and HCl solutions, what concentration are the solutions? Show a detailed description of the research method.
As a result, I will recommend the publication of this manuscript after major revision.
Round 2
Reviewer 1 Report
The manuscript has been revised according to my suggestions and comments. Furthermore, the authors provide sufficient information to the readers. I suggest this manuscript can be accepted in Coatings.
Minor editing of English language required by a native speaker.
Reviewer 2 Report
All the observations were satisfactorily addressed
Reviewer 3 Report
Figure 12 x-axis spelling needs to be corrected to Ultrasonication.
Once authors correct this the manuscript can be accepted.
Slight language corrections are necessary.
Reviewer 4 Report
Dear Author,
The additional result and discussion help to understand the common readers. Paper can now be accepted.